# Data-driven mathematical simulation analysis of emergency evacuation time in smart station's operations management

**Yang Hui** [1,2]*, **Qiang Yu**[3], **Hui Peng**[1]

**1** College of Transportation Engineering, Chang'an University, Xi'an, China, **2** School of Humanities, Chang'an University, Xi'an, China, **3** School of Automobile, Chang'an University, Xi'an, China

* huiyanghy@chd.edu.cn

## Abstract

This research establishes an emergency evacuation time model specifically designed for subway stations with complex structures. The model takes into account multiple factors, including passenger flow rate, subway facility parameters, and crowd density, to accurately assess evacuation times. It considers the impact of horizontal walking distance, flow rate, subway train size, and stair parameters on the overall evacuation process. By identifying bottleneck points such as gates, car doors, and stairs, the model facilitates the evaluation of evacuation capacity and the formulation of effective evacuation plans, particularly in multi-line subway transfer stations. The good consistency is achieved between the calculated evacuation time and simulated results using the Pathfinder software (with the relative error of 5.4%). To address urban traffic congestion and enhance subway station safety, the study recommends implemented measures for emergency diversion and passenger flow control. Additionally, the research presents characteristic mathematical models for various evacuation routes by considering the structural and temporal characteristics of metro systems. These models provide valuable guidance for conducting large-scale passenger evacuation simulations in complex environments. Future research can further enhance the model by incorporating psychological factors, evacuation signage, and strategies for vulnerable populations. Overall, this study contributes to a better understanding of evacuation dynamics and provides practical insights to improve safety and efficiency in subway systems.

## Introduction

As the process of urbanization accelerates, the demand for efficient transportation services have become increasingly urgent. To meet this demand, many cities have gradually established comprehensive transportation systems, encompassed road traffic, rail transit, and shared mobility, among others [1]. Among these modes of transportation, the subway has emerged as a preferred choice for citizens due to its reliability, high capacity, and efficient travel speed. However, the unique architectural features of metro stations, such as limited construction space, airtightness, restricted ventilation, and limited visibility, which give rise to a series of challenges especially for passenger evacuations, risks of congestion and stampedes [2].

**Data Availability Statement:** The paper does not involve specific passenger flow data. The data used in the simulation are derived from full passenger load conditions on a specific route and train. The corresponding emergency evacuation time is

calculated through mathematical modeling and simulation experiments.

**Funding:** This research was supported in part by the National Nature Science Foundation of China under Grant No. 52072044, in part by the National Science Foundation of Shaanxi Province under Grant No. 2021JQ-295. There was no additional external funding received for this study.

**Competing interests:** The authors have declared that no competing interests exist.

During emergencies, the structural characteristics of these subway stations can quickly exacerbate negative impacts. For instance, due to high passenger density, evacuating passengers may face difficulties in crowded spaces, leading to serious safety concerns. Additionally, the confined spaces and limited ventilation may result in the accumulation of toxic gases, intensifying the hazards during emergencies. In this context, accurately assessing the emergency evacuation and operational safety performance of subway stations have become crucial to ensure the safety of lives and property [3].

One key task among these assessments is the precise calculation of evacuation times in subway stations, determining whether they adhere to the guidelines outlined in the "Code for Design of Metro" (GB 50157-2013) [3]. This calculation serves not only as a significant indicator for evaluating the structural layout and evacuation capacity of subway stations but also as the foundation for formulating emergency evacuation plans and crowd control measures. However, traditional methods exhibit limitations in accurately predicting evacuation times due to the complexity and diversity of metro stations.

Hence, this research aims to address this issue by specifically designing an emergency evacuation time model, focusing on subway stations with complex structures. The model comprehensively considers multiple factors, including passenger flow rates, subway facility parameters, and crowd density, to achieve precise evaluations of evacuation times. By analyzing the influence of factors such as horizontal walking distance, flow rates, subway train sizes, and stair parameters on the overall evacuation process, the model can identify and quantify bottleneck points, such as gates, doors, and stairs, thereby facilitating the evaluation of evacuation capacity and the development of efficient evacuation plans, especially in complex situations such as multiline subway transfer stations. Overall, this research not only contributes to a deeper understanding of subway station evacuation dynamics but also offers practical insights to enhance safety and efficiency within subway systems. The development of a time model for emergency evacuations holds the potential to provide valuable guidance for the safety and sustainability of urban transportation systems.

## Literature review

Present domestic and overseas research mainly focuses on the mathematical modeling of crowd evacuation [4, 5], data fitting [6–8], simulation and testing [9–11], and areas that affect evacuation factors [12, 13].

Regarding the establishment of evacuation models, Chen et al. [14] utilized the M/G/c/c model to analyze the evacuation capacity of passages and stairs in subway stations, identifying them as the most congested bottlenecks. Chen et al. [15] proposed a shortest route algorithm based on fuzzy multifactor network weights, offering a theoretical derivation and mathematical calculation to demonstrate its practicality. Xu et al. [16] investigated passenger flow on subway platforms and quantitatively analyzed the boarding time of waiting passengers through mathematical modeling, resulting in a final calculation formula for boarding time. A comparison between experimental data and the mathematical model showed an error within an acceptable range of 10%. Although existed research has made progress in analyzing passenger and emergency evacuation flows in metro stations, the improvement in evaluating the impact of key facilities, obstacles, and overall evacuation processes is still lacking.

In the field of field investigation and data fitting, Togawa [17] proposed an empirical formula for the crowd evacuation time of complex buildings in the 1950s, which provided calculation results for comprehensive emergency evacuations based on different evacuee flow rates and exit widths. However, this formula overlooked the distribution of facilities inside the building, resulting in slightly shorter calculated evacuation times compared to actual values.

Dinenno [18] obtained reliable experimental data by conducting multiple evacuation drills in multistory buildings. They introduced the concept of "effective width" for stairs in their empirical formula for evacuation time. Parisi [19] introduced the "aspect area" in pedestrian advance and improved the social force model to better align with pedestrian evacuation situations. Zhou et al. [20] analyzed recent advancements in crowd evacuation guidance and comprehensively compared various methods such as static signs, dynamic signs, trained leaders, mobile devices, mobile robots, and wireless sensor networks. While these studies primarily focused on actual measurements and established empirical formula for single bottleneck areas, further research is needed to address the complexities of highly intricate scenes, such as transfer metro stations.

In the realm of simulation and testing research, Hu et al. [21] conducted simulations using Building Exodus software to consider crowd structure, the number of entrances and exits, stair widths, and fire conditions during evacuations. Mei et al. [22] developed a subway emergency evacuation simulation system with Pathfinder software, establishing quantitative relationships between evacuation time, the number of evacuated passengers, passenger flow rates, and other critical parameters. This system provided an objective foundation for science-based emergency evacuation management. Qin et al. [23] used Pathfinder to simulate the impact of different fire scenarios on passenger flows within subway stations. They found that stair entrances experienced the highest evacuation pressure, while exit widths had minimal effect on relieving congestion. Wu et al. [24] formulated an evacuation equilibrium bi-level programming model considering the travel time of pedestrian facilities with varying congestion levels. They designed an improved particle swarm optimization algorithm and simulated the evacuation process using Fuxingmen station on the Beijing subway as a case study. Kallianiotis et al. [25] employed Pathfinder to simulate passenger evacuation processes at a rail station, analyzing the influence of path selection and speed on evacuation time. Li et al. [26] employed Pathfinder simulation software to simulate passenger evacuation Processes at railway stations, identifying evacuation bottlenecks. Although these simulation tools adequately support the design of actual subway stations, they require detailed modeling and parameter settings for specific stations, resulting in low evaluation efficiency.

Regarding factors affecting evacuation, Dong et al [27] introduced agent technology into a cellular automata model and used Matlab software to simulate and analyze the impact of panic levels, emergency guidance, and other factors on evacuations. Jiang et al. [28] used Exodus building simulation software to model the evacuation processes of subway stations in 16 cases. They confirmed that adjusting stair widths and controlling the number of evacuees can reduce congestion and improve evacuation speed. Song et al. [29] discussed the influence of guides on evacuation efficiency based on factors such as their number, position, walking direction, and influence range. Li et al. [30] systematically analyzed the influence of various factors on passengers' psychological activities during emergency evacuations at subway stations, offering recommendations for guiding emergency evacuations based on these factors. Barron et al. [31] considered the characteristics of passengers' travel choice behavior during emergencies. Dell'olio et al. [32] analyzed passenger behavior under different emergency scenarios and proposed crowd flow guidance measures using the Spanish railway as an example. Zhou [33] comprehensively analyzed and compared crowd evacuation guidance methods such as static signs, dynamic signs, trained leaders, mobile devices, mobile robots, and wireless sensor networks. Hong [34, 35] developed a probability equilibrium model that considered changes in travel time reliability caused by congestion and introduced the concept of a travel time budget to expand the probability equilibrium model. Most studies have primarily analyzed the evacuation effects of individual passenger flows, the research on the evacuation effects in complex transfer stations and the influence of personnel behavior characteristics are limited.

While significant advancements have been made in evaluating the emergency evacuation capabilities of subway stations, research on emergency evacuation time has yet to consider the influence of key facilities and obstacles comprehensively. Curve fitting of evacuation time for key bottleneck areas based on empirical formulas and experimental data cannot be readily applied to specific evacuation scenarios, and limited research has been conducted on overall evacuation times. Additionally, various uncertainties in subway station emergency evacuations, including human, construction, environmental, traffic, and management factors, pose challenges in employing traditional empirical formulas and computer simulation methods.

To address these challenges, this paper comprehensively considers the unique geographical structures of subway stations and divides the evacuation process into segments based on critical nodes. The influence of key facilities and obstacles is analyzed and incorporated into an overall evacuation time model, accounting for multiple factors within each segment. The developed theoretical model in this paper is validated by comparing its calculated values with simulated values using the Pathfinder software. By integrating these aspects, this research aims to enhance understanding of emergency evacuation dynamics and provide practical insights for improving safety and efficiency in metro systems.

## Establishment of the evacuation time model

### Research methodology

This research adopts a segmented research approach to develop models for different time periods within subway station evacuations. Through on-site investigations and a comprehensive analysis of common factors in subway station facility structures, it is deduced that key facility structures along the evacuation route encompass the subway train, platform, platform stairs, station hall, gates, and exit stairs. The evacuation process is segmented into five stages, corresponding to distinct periods: ① from subway train to platform; ② from platform to stairway entrance; ③ from platform stairs to station hall (including stair congestion time and travel time on the stairs); ④ from station hall to stairway (including congestion time at gates); ⑤ from station hall stairs to ground floor (including stair congestion time and travel time on the stairs).

The formula for computing evacuation time is formulated based on the total number of individuals within the subway station (including those on the subway train, platform, and station hall), the rate of pedestrian flow, and population density. The iterative accumulation of time is carried out in accordance with the principle of continuous pedestrian movement. Subsequently, the model's outcomes are juxtaposed with the Pathfinder software's calculations for subway station evacuations, demonstrating the model's efficacy in elucidating crowd dynamics during emergency evacuations within subway stations. The technical roadmap for this study is depicted in Fig 1.

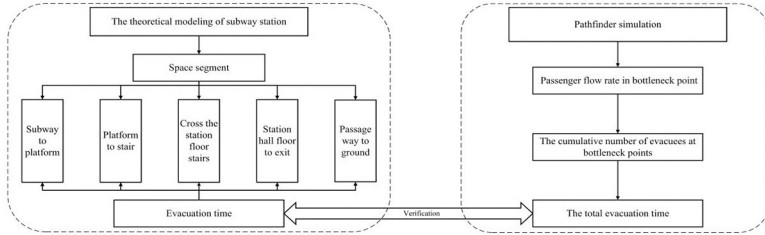

**Fig 1. The technical roadmap.**

## Model assumptions

The segmented evacuation time model based on multifactor analysis under the following specific assumptions:

(1) Negligible Stair Traffic: Prior to the commencement of evacuation, the presence of individuals on the stairs is minimal and poses no obstruction to the evacuation process.

(2) Initial Evacuation Flow: At the onset of evacuation, the model does not account for the influence of passenger flow adjustments within the subway stations.

(3) Average Distribution of Individuals: The distribution of people within the subway, platform, and station hall follows an average pattern. This implies that subjective factors influence the selection of each elevator and passage, and all stairways and passages possess identical capacities.

(4) Escalator Mode Change: At the outset of evacuation, automatic escalators are halted and function as inclined walkways (stairs).

(5) Station Hall Exit Gates: During emergency evacuation, the gates on the station hall floor are fully open, and ticket gates are employed as evacuation exits.

(6) Station Hall to Ground Floor Stairs: The model does not consider hindrances to pedestrians posed by inflection points on stairs connecting the station hall to the ground floor.

(7) Transfer Station Stairs: Stairs connecting two platforms within a transfer station are not designated as planned evacuation routes for emergency situations.

## Equations for calculating evacuation time

(1) $t_1$: Time to evacuate from the subway to the platform

Previous investigators have focused on analyzing the relationship between the number of passengers and the evacuation time from the perspective of mathematical models and experimental research. Based on this research, it has been concluded that on a horizontal platform, the time required for all passengers to exit the subway can be calculated using the following equation [36]:

$$t_1 = \frac{s^2 n^2}{2Kw} \cdot \frac{2.2 + \frac{e}{2}}{2.2 - \frac{e}{2}} \ln \frac{4.4}{e} \tag{1}$$

$$n = \frac{N}{f} \tag{2}$$

(2) $t_2$: Time to evacuate from the platform to the stairway entrance

The relationship between flow velocity and flow density was analyzed from a dynamics perspective, and it was concluded that the flow velocity in horizontal places satisfies the equations [37]:

$$v_0 = v_m(\alpha A + \beta B + \gamma) \tag{3}$$

$$A = 1.32 - 0.82 \ln \rho_i \tag{4}$$

$$B = 3.0 - 0.76 \rho_i \tag{5}$$

where $v_0$ refers to the horizontal flow rate of evacuated people on the platform floor in m/s, and $v_m$ is the maximum average flow rate in $v_m = 1.2$m/s. In an emergency, the maximum average flow rate. The variables $\alpha, \beta$ refer to the contributed weight to the flow rate under the influence of the front and rear, left and right, and other factors, respectively. The values of $\alpha$ are 0.25 and 0.44, $\beta$ is 0.014 and 0.088, and $\gamma$ is 0.15 and 0.26. The average values of $\alpha, \beta, \gamma$ were selected as the final weights: $\alpha = 0.35$, $\beta = 0.05$, and $\gamma = 0.20$ (rounded to two decimal places).

Time $t_2$ is computed as

$$t_2 = \frac{L}{v_0} \tag{6}$$

where $L$ is the maximum distance from the crowd to the stairway entrance in m.

(3) $t_3$: Time to travel the stairs from the platform

The time the crowd spends on the stairs is divided into two parts: the time for the crowd to travel the stairs and the congestion time on the platform floor at the stairway entrance.

Previous researchers [38] obtained the following expression for the time for everyone on the station floor to travel the stairs:

$$\frac{b}{m} = \frac{8.04}{t_{31}^{1.37}} \tag{7}$$

$$t_{31} = {}^{1.37}\sqrt{\frac{8.04m_1}{b_1}} \tag{8}$$

where $b_1$ refers to the effective width of the stairs on the platform floor, which is the width in m actually used by pedestrians, and $m_1$ is the total number of people on the platform floor in pers.

Time $t_{32}$ is the time needed for everyone on the platform floor to travel the stairs, calculated as

$$t_{32} = \frac{s_3}{v_3}\left(\frac{1.57r_1}{l_1}\right)^{\frac{1}{2}} \tag{9}$$

where $s_3$ is the horizontal stairway length on the platform floor in m, $r_1$ is the stair rung height on the platform floor in m, $l_1$ is the stair step height on the platform floor in m, and $v_3$ is people's stair-climbing speed in m/s.

(4) $t_4$: Time to evacuate from the station hall floor to the exit

Based on the formula for calculating the time to evacuate from the platform floor to the stairway entrance and considering the congestion time for passengers to pass through the gates on the station hall floor, the time to evacuate from the station hall floor to the exit is calculated as

$$t_4 = t_{41} + t_{42} \tag{10}$$

$$\begin{aligned} v_0 &= v_m(\alpha A + \beta B + \gamma) \\ A &= 1.32 - 0.82\ln\rho_i \\ B &= 3.0 - 0.76\rho_i \end{aligned} \tag{11}$$

Where the parameters and variables in Eq (9) are defined in the description of Eq (2) above,

and

$$t_{41} = \frac{s_4}{v_4} \tag{12}$$

$$t_{42} = 1.37\sqrt{\frac{8.04m_2'}{b_2}} \tag{13}$$

where $s_4$ is the maximum distance from the crowd to the safety exit in m, $b_2$ is the effective width of the gate, which is to the actual width used in m, $m_2$ is the total number of people on the station hall floor in pers, and $m_2'$ is the total number of people in the enclosed gate area in pers. The equation for $m_2'$ is

$$m_2' = n_3 \cdot \frac{s_3}{s_2} \tag{14}$$

where $s_2$ is the actual usable floor area of the station hall in m2, and $s_3$ is the usable enclosed gate area in m2.

(5) $t_5$: Time to evacuate from the exit to the ground floor

This evacuation time is calculated using a method similar to the time for stair congestion and stair travel. Previous investigators [38] obtained the following formula for the congestion time $t_{51}$ for everyone on the station hall floor to travel the exit stairs:

$$\frac{b}{m} = \frac{8.04}{t_{31}^{1.37}} \tag{15}$$

$$t_{51} = 1.37\sqrt{\frac{8.04m_3}{b_3}} \tag{16}$$

and the time for everyone on the station hall floor to travel the stairs is $t_{52}$, calculated as

$$t_{52} = \frac{s_5}{v_5}\left(\frac{1.57r}{l}\right)^{\frac{1}{2}} \tag{17}$$

where the description of Eq (3) above provides the parameter descriptions for Eq (15).

(6) Additional equations used in the calculations are

Maximum evacuation capacity of the car doors:

$$q_1 = \frac{n_1}{t_1} \tag{18}$$

Maximum evacuation capacity of platform stairs:

$$q_1 = \frac{n_3}{t_3} \tag{19}$$

Maximum evacuation capacity of station hall stairs:

$$q_1 = \frac{n_3}{t_5} \tag{20}$$

Maximum evacuation capacity of the station hall floor (including turnstiles):

$$q_4 = \frac{n_3}{t_4} \cdot \frac{s_3}{s_2} \tag{21}$$

Based on the principle that the person closest to the exit (the first person to the ground) is safely evacuated first, the time for everyone in the subway station to reach the safety zone is taken as the total evacuation time $t_{total}$:

$$t_{extra} = [(t_4 + t_5 - t_2) \cdot q_2 + (t_4 + t_5 - t_1 - t_2) \cdot q_2]/q_3 \tag{22}$$

$$t_{total} = [t_4 + t_5] + t_4 + t_{extra} \tag{23}$$

where $t_{extra}$ is the time needed for the remaining people on the platform floor and the subway train to travel the exit stairs after the people on the station hall floor are evacuated, $t_4 + t_5$ is the time for the people on the station hall floor to be evacuated, and $t_4$ is the time for the people on the platform floor and the subway train to pass through the station hall.

## Analysis of experimental data and simulation results: A case study of evacuation time model

This section validates established evacuation time model through data analysis and simulations, demonstrating its real-world applicability in a specific subway station case study. The intricate station layout, featuring platforms, station halls, stairs, and exits, underscores emergency challenges. Visual representations lay the groundwork for thorough analysis and simulations, examining dynamics, bottlenecks, and evacuation scenarios.

### Research case statement and visualization of the station layout

This case study centers on the Wulukou subway station located in Xi'an, China, as the focal point of this case study. The aim of this study is to employ our established model to calculate evacuation times and analyze passenger evacuation pathways within the subway station. The selected subway station serves as a pivotal transfer point for both metro Line 1 and Line 4, encompassing the platform areas of both lines, as well as the station hall floor which comprises both paid and non-paid sections. Specifically, the platform areas for Line 4 and Line 1 measure 1427 m2 and 1410 m2 respectively. The paid section of the station hall occupies an area of 1344 m2, with the non-paid area extending over 3380 m2. These spatial distributions are visually represented in Figs 2–4. In order to conduct emergency evacuation simulations, the Pathfinder software in conjunction with computer-aided design (CAD) techniques is normally utilized to meticulously craft the simulation environment.

### Analysis of theoretical model values and evacuation time calculation

Evacuation time is calculated based on the established mathematical model by taking into account the actual conditions of Wulukou metro station. The distribution of evacuation facilities in the station includes stairs A, B, C, and D, as well as exits A, B, C, D, E, and F. Considering the specific characteristics of Wulukou subway station, the number of evacuees and other relevant parameters are determined for the calculation.

For the number of evacuees, the vehicle in Wulukou Subway Station is type B according to the real-time passenger flow and the particular geographical location of Wulukou Subway Station. Each car occupies an area of 52 m2. Considering the area occupied by the seats on both

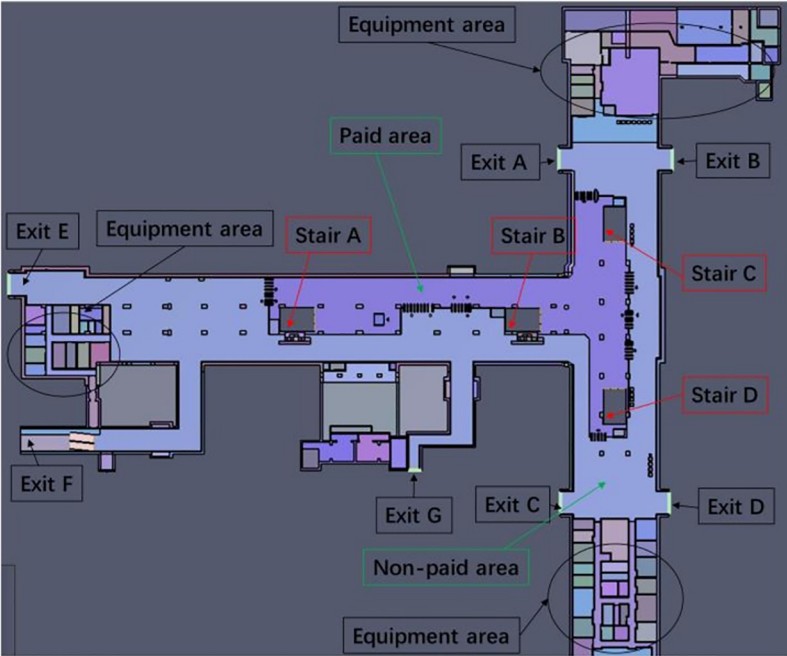

**Fig 2. Schematic diagram of the station hall floor.**

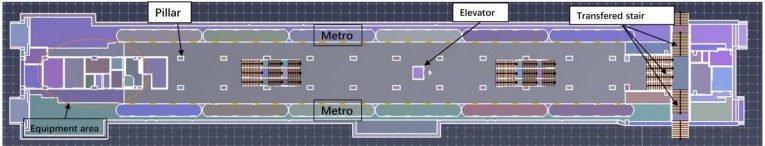

**Fig 3. Schematic diagram of the Line 4 platform floor.**

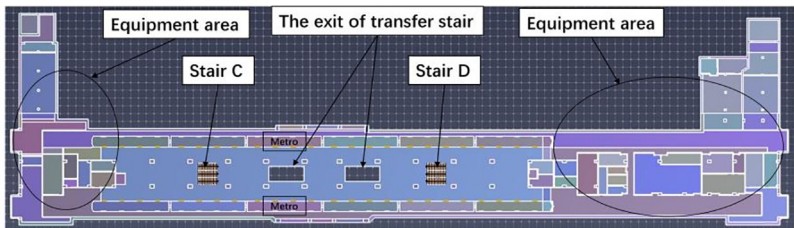

**Fig 4. Schematic diagram of the Line 1 platform floor.**

sides and the anti-fall handrails in the middle, the actual area of each car is 40 m$^2$. There are six cars, and the platform is an island-style structure. During morning rush hours, the average density of passengers on the subway train is 3 pers/m$^2$, during which passengers would be in contact with each other. Therefore, the total number of passengers in a train is $N_0 = 40 \times 6 \times 3 = 720$ pers.

**Table 1. Theoretical model results.**

| Category | Line 4 | Line 1 |
|---|---|---|
| Number of people in a train (pers) | 1440 | 1440 |
| Number of people on the platform floor (pers) | 285 | 282 |
| Number of people on the station hall floor (pers) | 607 | 607 |
| Evacuation capacity of the car doors (pers/sec) | 8.58 | 8.57 |
| Evacuation capacity of platform stairs (pers/sec) | 9.39 | 12.99 |
| Evacuation capacity of station hall stairs (pers/sec) | 11.3 | 12.45 |
| Evacuation capacity of the turnstile (pers/sec) | 1.26 | 1.26 |
| Time to exit the train (sec) | 167.87 | 167.87 |
| Time on the platform (sec) | 30.35 | 21.72 |
| Time on the stairs (sec) | 53.72 | 48.8 |
| Time on the station hall (sec) | 120.52 | 120.52 |
| Time of exit (sec) | 87.74 | 87.74 |
| Total time (sec) | 341.92 | 291.97 |

In the station hall area, considering the field data analysis, the number of people in the non-paid area is 338 pers, while the number in the paid area is 269 pers. The number of people on the Line 4 platform floor is 285 pers, while there are 282 pers on the Line 1 platform floor. According to the simulated evacuation conditions, with four trains arriving at the station simultaneously and the distribution of evacuees on the platform floor and the station hall floor unchanged, the total number of evacuees was 4054 pers.

According to the values in Eq (3), $\alpha = 0.35$, $\beta = 0.05$, and $\gamma = 0.20$. Under emergency evacuation conditions [39], the maximum average flow rate of passengers is 1.2 m/s in Eq (3). Thus, the flow rate on the platform floor is obtained as 1.19 m/s. Survey data is analyzed according to the actual conditions, in which the step height of the stairs is $r = 0.18$ m, and the step width is $b = 0.28$ m. The horizontal length of the stairs is $L = 27.65$ m for Line 4 and $L = 10.80$ m for Line 1. The effective width of the four sets of stairs is $1.9 \times 3 \times 4$ m, the maximum distance from the Line 4 platform floor to the stairway entrance is 46 m, and the maximum distance from the Line 1 platform floor to the stairway entrance is 33 m.

Considering the particular geographical structure of Wulukou Subway Station, the cumulative evacuation time of the platform floor of Line 4 and Line 1 is first calculated as $T_1$ and $T_2$. The final total evacuation time is

$$T_{total} = \max \{T_1, T_2\} \tag{24}$$

The established mathematical model is solved using the GUI function module of MATLAB [40], and the results are shown in Table 1.

## Simulation experiment research and comparative analysis using pathfinder

Utilizing the Pathfinder software [41], a simulation of passenger flow at Xi'an Wulukou Metro Station under extreme emergency evacuation scenarios was conducted. Pathfinder, an agent-based evacuation simulation software developed by Thunderhead Engineering Company (USA), offers two distinct modes of movement simulation: the Society of Fire Protection Engineers (SFPE) mode and the steering mode.

In the SFPE mode, the evacuation route is determined primarily by walking route length, whereby passengers opt for the nearest exit in terms of proximity. This mode automatically

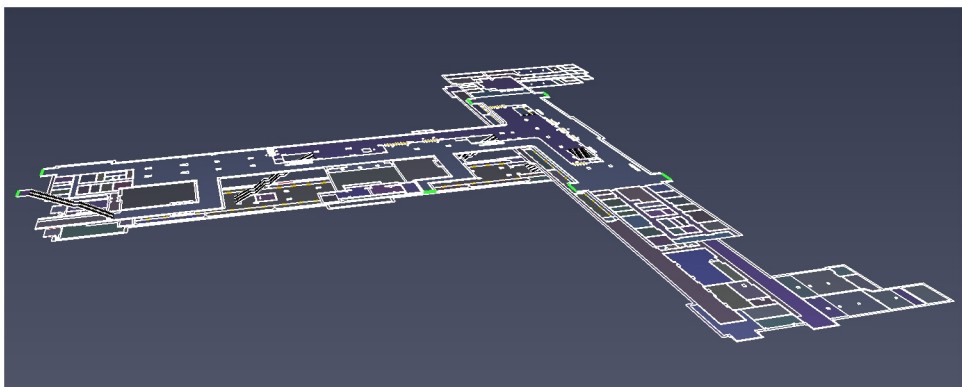

**Fig 5. Three-dimensional view of the Wulukou subway station.**

adjusts the passenger flow rate by gauging evacuation space density and the passenger flow is restricted by doors.

Conversely, the steering mode considers factors like route planning and passenger interactions. Evacuation routes are established based on evacuation distances and passenger proximity. In this mode, doors no longer act as flow restrictors, allowing passengers to complete ongoing movements and react to changing environmental conditions. Given its practical applicability to real-life situations, the steering mode was chosen for the simulation experiment in this study.

Fig 5 presents a comprehensive three-dimensional depiction of Wulukou Subway Station, showcasing the spatial arrangement of the Line 4 platform floor, the Line 1 platform floor, and the entirety of the station hall floor spanning both Line 1 and Line 4. The station's layout assumes a distinctive T-shaped configuration, providing a clear overview of the architectural arrangement. Passenger distribution inside the subway station are shown in Table 2.

Subsequently, Fig 6 offers insights into the distribution of passengers after an elapsed evacuation period of 50 seconds. Evidently, within this timeframe, passengers on both the platform floor and the station hall floor converge towards the stairway entrances, resulting in the emergence of a congestion point. Furthermore, passengers situated on the station hall floor exhibit a distinct gathering pattern around the gates, forming yet another bottleneck in the evacuation process.

These visual representations provide a tangible illustration of the evolving dynamics within the station during emergency evacuation scenarios. The dynamic inter-play between passenger movements, bottlenecks, and congestion points serves as a valuable reference for

**Table 2. Passenger distribution inside the subway station.**

| Subway station structure | Inner structure | Usable area (m²) | Number of people |
|---|---|---|---|
| Platform floor | Platform floor of Line 1 | 1410 | 282 |
| | Platform floor of Line 4 | 1427 | 285 |
| | Train on the left side of Line 4 | 312 | 720 |
| | Train on the right side of Line 4 | 312 | 720 |
| | Train on the left side of Line 1 | 312 | 720 |
| | Train on the right side of Line 1 | 312 | 720 |
| Station hall | Paid area | 1344 | 269 |
| | Non-paid area | 3380 | 338 |

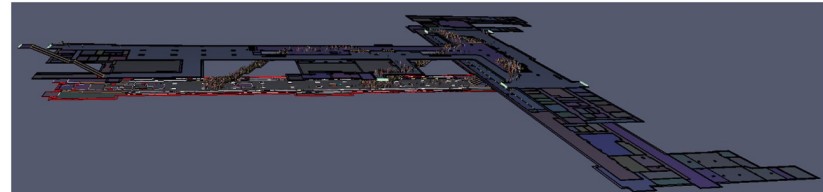

**Fig 6. The distribution of passengers after 50 seconds of evacuation time.**

understanding the intricacies of the evacuation process and underscores the significance of efficient route planning and bottleneck mitigation strategies.

Fig 7(a) is the distribution diagram of passenger flow on the subway system stairs, and Fig 7(c) is the distribution diagram of the cumulative evacuees in the subway system. Based on the analysis results, it can be concluded that the overall evacuation capacity has been improved. During the evacuation, the cumulative number of evacuees on the left stairs of C and D reaches their maximum values (474 and 478, respectively). The flow rate is recorded as $1.5 \sim 2.5$ pers/s, and the utilization rate increases as some passengers pass through the transfer station, traveling from the Line 4 platform floor to the Line 1 platform floor. According to the proximity principle, passengers prioritize the surrounding stairs (the left stairs of C and D). In this situation, the utilization rate of stairs A and B is lower than for stairs C and D because congestion is less likely to occur in Line 4 based on its relatively long walking distance on the platform floor and low passenger density.

Fig 7(b) is the distribution diagram of the passenger flow at the exits, and Fig 7(d) is the distribution diagram of the cumulative evacuees at the subway system exits. The passenger flow rate of Exit A is the largest throughout the evacuation at $3 \sim 5$ pers/s, and the cumulative number of evacuees is as high as 1,180. Further analysis also reveals that the gates close to Exit A are subject to more severe congestion, prolonging evacuation time. The Exits B, C, and G utilization rates are relatively high, with the cumulative number of evacuees at 686, 829, and 746, respectively. Exits E and F have the lowest utilization rates, with 163 and 23 cumulative evacuees.

The highest passenger flows during the $50 \sim 60$ s period are recorded for Exits E and F. However, they are idle with zero flow after 60 s. Therefore, guides could be added to relieve the

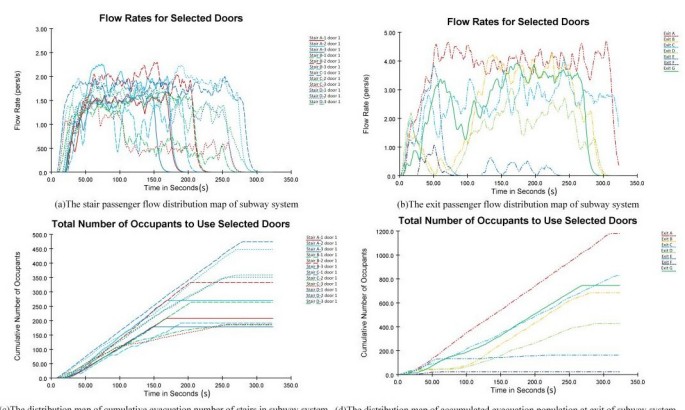

(a)The stair passenger flow distribution map of subway system    (b)The exit passenger flow distribution map of subway system

(c)The distribution map of cumulative evacuation number of stairs in subway system    (d)The distribution map of accumulated evacuation population at exit of subway system

**Fig 7. Evacuee distribution maps of bottleneck points at the subway station.**

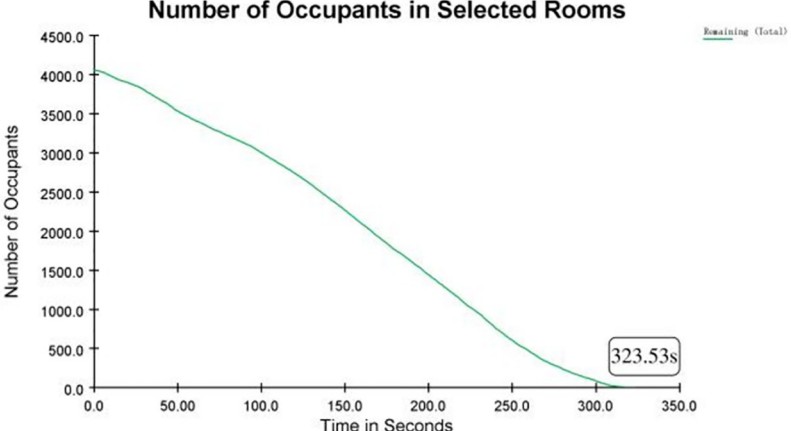

**Fig 8. The relationship between remaining evacuees and required evacuation time.**

pressure of bottleneck points and improve evacuation efficiency. Fig 8 shows that it would take 323.53 s to evacuate all the passengers at Wulukou Subway Station.

In this simulation, the error between the calculated theoretical evacuation time and the value of the simulation experiment is

$$\varepsilon = \frac{|t_{theoretical} - t_{simulation}|}{t_{theoretical}} \times 100\% = \frac{|341.92 - 323.53|}{341.92} \times 100\% = 5.4\% \tag{25}$$

The high degree of consistency between the results obtained from theoretical model and simulation experiment is achieved, which provides substantial validation of the effectiveness of the established mathematical model. The calculated evacuation time of 341.92 seconds is in good agreement with the predictions of the mathematical model with an error rate of 5.4% ($< 10\%$). This outcome not only attests to the credibility of the mathematical model but also underscores the precision of the model in forecasting emergency evacuation times.

## Conclusions

The conclusions from this study are summarized as follows:

(1) An emergency evacuation time model was developed for subway stations with complex structures by taking into account factors such as passenger flow rate, subway facility parameters, and crowd density. The model identified horizontal walking distance, flow rate, subway train size, and stair parameters as the main factors influencing evacuation time.

(2) The emergency evacuation model can predict the locations of bottleneck points. These bottleneck points are the gates > car doors > stairs (in descending order). "Arch-shaped" congestion is most likely to occur at the gates. The model provides a foundation for evaluating the emergency evacuation capacity of multiline sub-way transfer stations and defines as an effective reference for formulating emergency evacuation plans.

(3) The calculated evacuation time from the mathematical model closely aligned with the results obtained from the simulation experiment using the Pathfinder software, with an error rate of only 5.4%. This demonstrates the scientific validity and reliability of the emergency evacuation model proposed in this study.

To address issues of urban traffic congestion and enhance subway station safety, effective measures for emergency diversion and passenger flow control are recommended. Considering the structural and temporal characteristics of subway systems, this study introduced characteristic mathematical models for different evacuation routes, encompassing trains, platforms, stairs, gates, and station halls. By incorporating real-time changes in passenger flow, the total time evacuation model was derived re-cursively, enhancing calculation efficiency. The model offers theoretical and practical guidance for simulating large-scale passenger evacuations in complex environments.

Future research can expand upon the model by considering psychological factors affecting evacuees, evacuation signage at stations, and evacuation strategies for vulnerable populations. The added factors would further enhance the model's comprehensiveness, practicality, and scientific rigor.

In summary, this study establishes an emergency evacuation time model for subway stations, providing insights into evacuation dynamics and aiding in the development of strategies to improve safety and efficiency in subway systems.

## Author Contributions

**Conceptualization:** Yang Hui, Qiang Yu.

**Data curation:** Yang Hui.

**Formal analysis:** Qiang Yu.

**Funding acquisition:** Yang Hui, Hui Peng.

**Investigation:** Hui Peng.

**Methodology:** Yang Hui, Qiang Yu, Hui Peng.

**Project administration:** Hui Peng.

**Resources:** Qiang Yu.

**Software:** Yang Hui.

**Visualization:** Yang Hui.

**Writing – original draft:** Yang Hui.

**Writing – review & editing:** Yang Hui.

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
