## [Decision Letter · Decision Letter 0]

20 Nov 2023

PONE-D-23-32821Data-driven Mathematical Simulation Analysis of Emergency Evacuation Time in Smart Station’s Operation Management.PLOS ONE

Dear Dr. Hui,

Thank you for submitting your manuscript to PLOS ONE. After careful consideration, we feel that it has merit but does not fully meet PLOS ONE’s publication criteria as it currently stands. Therefore, we invite you to submit a revised version of the manuscript that addresses the points raised during the review process.Please consider all comments  Please submit your revised manuscript by Jan 04 2024 11:59PM. If you will need more time than this to complete your revisions, please reply to this message or contact the journal office at plosone@plos.org. Please include the following items when submitting your revised manuscript:A rebuttal letter that responds to each point raised by the academic editor and reviewer(s). You should upload this letter as a separate file labeled 'Response to Reviewers'.A marked-up copy of your manuscript that highlights changes made to the original version. You should upload this as a separate file labeled 'Revised Manuscript with Track Changes'.An unmarked version of your revised paper without tracked changes. You should upload this as a separate file labeled 'Manuscript'.

We look forward to receiving your revised manuscript.

Kind regards,

Ahmed Mancy Mosa, Ph.D.

Academic Editor

PLOS ONE

Journal Requirements:

This research was supported in part by the National Nature Science Foundation of China under Grant No. 52072044, in part by the National Science Foundation of Shaanxi Province under Grant No. 2021JQ-295.

7. Please amend either the title on the online submission form (via Edit Submission) or the title in the manuscript so that they are identical.

Reviewers' comments:

Reviewer's Responses to Questions

**Comments to the Author**

1. Is the manuscript technically sound, and do the data support the conclusions?

Reviewer #1: Partly

Reviewer #2: Yes

2. Has the statistical analysis been performed appropriately and rigorously? 

Reviewer #1: I Don't Know

Reviewer #2: Yes

3. Have the authors made all data underlying the findings in their manuscript fully available?

Reviewer #1: No

Reviewer #2: Yes

4. Is the manuscript presented in an intelligible fashion and written in standard English?

Reviewer #1: Yes

Reviewer #2: Yes

5. Review Comments to the Author

Reviewer #1: The research proposes a model for estimating evacuation time, which has been verified using the Pathfinder software. However, the paper lacks clarity regarding the specific formula that constitutes its primary contribution. The Eq(1) and (2) for the time required for evacuation from the subway to the platform, and if this is a contribution from reference 36. Similarly, the evacuation time from the platform to the stairway entrance appears to be linked to reference 37, and the time to traverse the stairs to reference 38. I think these things should be included in the "The technical roadmap for

this study is depicted in Fig 1.". It is hard to determine the suitability of the work for acceptance in PLOS ONE at its current stage.

Reviewer #2: This study contributes to a better understanding of evacuation dynamics and provides practical insights to improve safety and

efficiency in subway systems. The introduction provides a good overview of the problem and the motivation behind the research. The manuscript is well presented however only few details need to improve:

1.The methodology is adequately explained, but it would benefit from more details on the specific steps and calculations involved.

2. Include the data availability statement in the revised manuscript. The present manuscript does not explicitly state whether the authors have made all the data underlying the findings fully available.

6. PLOS authors have the option to publish the peer review history of their article (what does this mean?). If published, this will include your full peer review and any attached files.

Reviewer #1: No

Reviewer #2: No

---

## [Author Response · Author response to Decision Letter 0]

5 Jan 2024

Dear Editor and Reviewers: 

Thank you very much for giving us an opportunity to revise our manuscript ID PONE-D-23-32821. We appreciate for the positive and constructive comments and suggestions to improve the present manuscript. We have revised the manuscript thoroughly as per the comments and the concerns raised by all reviewers. All the changes made in the revision are highlighted with the blue-color text. Also, we have incorporated comments given in the Reviewer Attachment.

The details of responses to reviewers’ comments and additional questions are listed as follows:

[Comment 1]

1. Is the manuscript technically sound, and do the data support the conclusions?

Reviewer #1: Partly

Reviewer #2: Yes

[Response 1]

 Thank you very much for your advice.

As for Reviewer #1: In this study, the simulated data were normally adopted to validate the validity of the theoretical model. The calculated value and modeled value were in good agreement. In addition, the passenger density, walking speed and metro equipment were considered in theoretical calculation and numerical simulation to evaluate the needed evacuation time. Therefore, the obtained conclusions related to passenger factors (such as passenger flow rate, subway facility parameters and crowd density), facility factors (such as gates, car doors and stairs) and evacuation time are scientific and reasonable.

[Comment 2]

2. Has the statistical analysis been performed appropriately and rigorously?

Reviewer #1: I Don't Know

Reviewer #2: Yes

[Response 2]

I’m appreciated for your advice. 

As for Reviewer #1: This study established an emergency evacuation time model specifically designed for subway stations with complex structures. The model considered multiple factors, including passenger flow rate, subway facility parameters, and crowd density, to accurately assess evacuation times. It also investigates the effect of horizontal walking distance, flow rate, subway train size and stair parameters on the overall evacuation process. On the one hand, the passenger walking speed, subway facilities and passenger density were studied to investigate their effects on the total evacuation time. For another, the effect of stairs and exits on passenger flow rate and accumulated evacuation passengers were also analyzed. Finally, the calculated evacuation time is in good agreement with simulated evacuation time totally. Therefore, the statistical analysis in this paper is appropriate and rigorous based on the above-mentioned research logicality. 

[Comment 3]

3. Have the authors made all data underlying the findings in their manuscript fully available?

Reviewer #1: No

Reviewer #2: Yes

[Response 3]

It is my pleasure having your comment.

As for Reviewer #1: The proposed passenger number of evacuations are obtained based on the previous research foundation(such as passenger density, walking speed and equipment parameters at emergence situation). In this case, this paper simulates the maximum evacuated passengers of four trains at the transfer station and the maximum capacity of the station hall and platform floor. Such efforts are fully consistent with the number of passengers in a real emergency evacuation situation. In addition, it can also reasonably calculate the maximum evacuation capacity of different platform structures based on the existing crowd flow pattern. Therefore, the obtained data and findings in this study are fully available.

[Comment 4]

4. Is the manuscript presented in an intelligible fashion and written in standard English? PLOS ONE does not copyedit accepted manuscripts, so the language in submitted articles must be clear, correct, and unambiguous. Any typographical or grammatical errors should be corrected at revision, so please note any specific errors here.

Reviewer #1: Yes

Reviewer #2: Yes

[Response 4]

Thank you for your suggestions. To make the response and changes easier to identify where necessary, the editor’s and reviewers’ comments are presented with the response from author is presented in red, and the corresponding corrections in the revised manuscript are highlighted in blue.

In this study, we have changed the “subway” into “metro” to standardize the usage of words.

(1) Lines 13-14 page 1:

We have changed the “develops” into “establishes” to standardize the usage of words.

(2) Lines 20-22 page 1: 

The calculated evacuation time from the mathematical model closely aligns with simulation results obtained using the Pathfinder software, confirming its scientific validity with an error rate of 5.4%.

The sentence is revised as follows:

The good consistency is achieved between the calculated evacuation time and simulated results using the Pathfinder software (with the relative error of 5.4%).

(3) Lines 22-23 page 1:

We have changed the “implementing” into “implemented” to standardize the usage of words.

(4) Lines 24-26 page 1:

Additionally, the research presents characteristic mathematical models for various evacuation routes, taking into consideration the structural and temporal characteristics of metro systems.

The sentence is revised as follows:

Additionally, the research presents characteristic mathematical models for various evacuation routes by considering the structural and temporal characteristics of metro systems.

(5) Lines 39 page 1:

We have changed the “encompassing” into “encompassed” to standardize the usage of words.

(6) Lines 42-46 page 2:

However, the unique architectural features of metro stations, such as limited construction space, airtightness, restricted ventilation, and limited visibility, give rise to a series of challenges during emergency situations, particularly passenger evacuations, including risks of congestion and stampedes.

The sentence is revised as follows:

However, the unique architectural features of metro stations, such as limited construction space, airtightness, restricted ventilation, and limited visibility, which give rise to a series of challenges especially for passenger evacuations, risks of congestion and stampedes.

(7) Lines 59-60 page 1:

However, due to the complexity and diversity of metro stations, traditional methods exhibit limitations in accurately predicting evacuation times.

The sentence is revised as follows:

However, traditional methods exhibit limitations in accurately predicting evacuation times due to the complexity and diversity of metro stations.

(8) Lines 73-74 page 2:

We have changed the “development” into “developed” to standardize the usage of words.

(9) Lines 89-91 page 1-2:

Although existing research has made progress in analyzing passenger and emergency evacuation flows in metro stations, there is still room for improvement in evaluating the impact of key facilities, obstacles, and overall evacuation processes.

The sentence is revised as follows:

Although existed research has made progress in analyzing passenger and emergency evacuation flows in metro stations, the improvement in evaluating the impact of key facilities, obstacles, and overall evacuation processes is still lacking. 

(10) Lines 106-109 page 3:

While these studies primarily focused on actual measurements and empirical formula construction for single bottleneck areas, further research is needed to address the complexities of highly intricate scenes, such as metro stations.

The sentence is revised as follows:

While these studies primarily focused on actual measurements and established empirical formula for single bottleneck areas, further research is needed to address the complexities of highly intricate scenes, such as transfer metro stations.

(11) Lines 147-150 page 3-4:

Most studies have primarily analyzed the evacuation effects of individual passenger flows, with limited research on the evacuation effects in complex transfer stations and the influence of personnel behavior characteristics.

The sentence is revised as follows:

Most studies have primarily analyzed the evacuation effects of individual passenger flows, the research on the evacuation effects in complex transfer stations and the influence of personnel behavior characteristics are limited.

(12) Lines 164-168 page 4:

The theoretical model developed in this paper is validated by comparing its results with simulations conducted using the Pathfinder software. By integrating these aspects, this research aims to enhance our understanding of emergency evacuation dynamics and provide practical insights for improving safety and efficiency in metro systems.

The sentence is revised as follows:

The developed theoretical model in this paper is validated by comparing its calculated values with simulated values using the Pathfinder software. By integrating these aspects, this research aims to enhance understanding of emergency evacuation dynamics and provide practical insights for improving safety and efficiency in metro systems.

(13) Lines 190-191 page 5:

The segmented evacuation time model, developed through multifactor analysis, operates under the following specific assumptions:

The sentence is revised as follows:

The segmented evacuation time model based on multifactor analysis under the following specific assumptions:

(14) Lines 313-314 page 8:

We have changed the “our” into “established” to standardize the usage of words.

(15) Lines 320-321 page 8:

We have changed the “our” into “this” to standardize the usage of words.

(16) Lines 328-329 page 8:

we utilize the Pathfinder software in conjunction with computer-aided design (CAD) techniques to meticulously craft the simulation environment.

The sentence is revised as follows:

The Pathfinder software in conjunction with computer-aided design (CAD) techniques is normally utilized to meticulously craft the simulation environment.

(17) Lines 340-341 page 9:

Evacuation time is calculated based on the established mathematical model, taking into account the actual conditions of Wulukou metro station.

The sentence is revised as follows:

Evacuation time is calculated based on the established mathematical model by taking into account the actual conditions of Wulukou metro station.

(18) Lines 380-381 page 11:

we conducted a simulation of passenger flow at Xi’an Wulukou Metro Station under extreme emergency evacuation scenarios

The sentence is revised as follows:

A simulation of passenger flow at Xi’an Wulukou Metro Station under extreme emergency evacuation scenarios was conducted

(19) Lines 386-388 page 11:

This mode automatically adjusts the passenger flow rate by gauging evacuation space density, with doors acting as flow-restricting elements.

The sentence is revised as follows:

This mode automatically adjusts the passenger flow rate by gauging evacuation space density and the passenger flow is restricted by doors.

(20) Lines 420-423 page 12:

In this situation, the utilization rate of stairs A and B is lower than for stairs C and D because congestion is less likely to occur in Line 4 because of its relatively long walking distance on the platform floor and low passenger density.

The sentence is revised as follows:

In this situation, the utilization rate of stairs A and B is lower than for stairs C and D because congestion is less likely to occur in Line 4 based on its relatively long walking distance on the platform floor and low passenger density.

(21) Lines 441-443 page 13:

The high degree of consistency between the results obtained from the emergency evacuation simulation experiment and the theoretical model provides substantial validation of the effectiveness of the established mathematical model.

The sentence is revised as follows:

The high degree of consistency between the results obtained from theoretical model and simulation experiment is achieved, which provides substantial validation of the effectiveness of the established mathematical model.

(22) Lines 443-444 page 13:

The calculated evacuation time of 341.92 seconds, as derived from the research, aligns closely with the predictions of the mathematical model, with an error rate of 5.4% that falls below the 10% threshold.

The sentence is revised as follows:

The calculated evacuation time of 341.92 seconds is in good agreement with the predictions of the mathematical model with an error rate of 5.4 % (<10 %).

(23) Lines 456 page 14:

We have changed the “is” into “defines as” to standardize the usage of words.

(24) Lines 470-471 page 14:

These additions would further enhance the model's comprehensiveness, practicality, and scientific rigor.

The sentence is revised as follows:

The added factors would further enhance the model's comprehensiveness, practicality, and scientific rigor.

[Comment 5]

5. Review Comments to the Author

Reviewer #1: This research proposes a model for estimating evacuation time, which has been verified using the Pathfinder software. However, the paper lacks clarity regarding the specific formula that constitutes its primary contribution. The Eq(1) and (2) for the time required for evacuation from the subway to the platform, and if this is a contribution from reference 36. Similarly, the evacuation time from the platform to the stairway entrance appears to be linked to reference 37, and the time to traverse the stairs to reference 38. I think these things should be included in the "The technical roadmap for this study is depicted in Fig 1.". It is hard to determine the suitability of the work for acceptance in PLOS ONE at its current stage.

Reviewer #2: This study contributes to a better understanding of evacuation dynamics and provides practical insights to improve safety and efficiency in subway systems. The introduction provides a good overview of the problem and the motivation behind the research. The manuscript is well presented however only few details need to improve:

 1. The methodology is adequately explained, but it would benefit from more details on the specific steps and calculations involved.

 2. Include the data availability statement in the revised manuscript. The present manuscript does not explicitly state whether the authors have made all the data underlying the findings fully available.

[Response 5]

Thank you very much for your advice.

As for Reviewer #1: The evacuation process is divided into five stages, corresponding to distinct periods: ① from subway train to platform; ② from platform to stairway entrance; ③ from platform stairs to station hall (including stair congestion time and travel time on the stairs); ④ from station hall to stairway (including congestion time at gates); ⑤ from station hall stairs to ground floor (including stair congestion time and travel time on the stairs). As for different evacuation stage, the corresponding evacuation model is established based on the previous researches and findings (Such as Refs. [36]-[38]). Generally, passenger evacuation at metro station is mainly included the above-mentioned five stages. The built numerical models can apply single platform and a transfer station with multiple intersected routes. Therefore, the established evacuation models are universal and applicable.

As for Reviewer #2:

Aiming at Q1, the GUI interface developed by MATLAB version 2022a is obtained by importing the relevant calculation models at each evacuation stage. This method has a high solution efficiency and precision. Additionally, the total evacuation time is our research object due to the validation with numerical modeling in the subsequent chapters. Therefore, the utilized solution in this study is reasonable and acceptable. 

Aiming at Q2, in this study, this paper simulates the evacuated people at a subway transfer station under the condition of large passenger flow. The proposed passengers have made clear and specific statement in Analysis of Theoretical Model Values and Evacuation Time Calculation Part based on the reasonable analysis and previous researches foundation. Therefore, the available data is scientific and significant. 

[Comment 6]

6. PLOS authors have the option to publish the peer review history of their article (what does this mean?). If published, this will include your full peer review and any attached files.

 Do you want your identity to be public for this peer review? For information about this choice, including consent withdrawal, please see our Privacy Policy.

Reviewer #1: No

Reviewer #2: No

[Response 6]

Thank you for your valuable suggestions. The relevant comments from the editor and reviewers were carefully modified point to point. Please see the attached file. We deeply appreciate your consideration of our manuscript and look forward to your suggestions on this revised manuscript. Please don’t hesitate to contact us if you have any questions.

---

## [Decision Letter · Decision Letter 1]

29 Jan 2024

Data-driven Mathematical Simulation Analysis of Emergency Evacuation Time in Smart Station’s Operation Management.

PONE-D-23-32821R1

Dear Dr. Hui,

We’re pleased to inform you that your manuscript has been judged scientifically suitable for publication and will be formally accepted for publication once it meets all outstanding technical requirements.

Kind regards,

Ahmed Mancy Mosa, Ph.D.

Academic Editor

PLOS ONE

Additional Editor Comments (optional):

Reviewers' comments:

Reviewer's Responses to Questions

**Comments to the Author**

1. If the authors have adequately addressed your comments raised in a previous round of review and you feel that this manuscript is now acceptable for publication, you may indicate that here to bypass the “Comments to the Author” section, enter your conflict of interest statement in the “Confidential to Editor” section, and submit your "Accept" recommendation.

Reviewer #1: All comments have been addressed

2. Is the manuscript technically sound, and do the data support the conclusions?

Reviewer #1: Yes

3. Has the statistical analysis been performed appropriately and rigorously? 

Reviewer #1: N/A

4. Have the authors made all data underlying the findings in their manuscript fully available?

Reviewer #1: Yes

5. Is the manuscript presented in an intelligible fashion and written in standard English?

Reviewer #1: Yes

6. Review Comments to the Author

Reviewer #1: The author revised the manuscript with more clarity than the last time. I think the manuscript is suitable to publish in PLOS ONE.

7. PLOS authors have the option to publish the peer review history of their article (what does this mean?). If published, this will include your full peer review and any attached files.

Reviewer #1: No

---

## [Editor Report · Acceptance letter]

8 Feb 2024

PONE-D-23-32821R1 

PLOS ONE

Dear Dr. Hui, 

I'm pleased to inform you that your manuscript has been deemed suitable for publication in PLOS ONE. Congratulations! Your manuscript is now being handed over to our production team.

Kind regards, 

on behalf of

Dr. Ahmed Mancy Mosa 

Academic Editor

PLOS ONE